# The Effect of Mg, Fe(II), and Al Doping on CH_4_: Adsorption and Diffusion on the Surface of Na-Kaolinite (001) by Molecular Simulations

**DOI:** 10.3390/molecules25041001

**Published:** 2020-02-24

**Authors:** Kai Wang, Bin Zhang, Tianhe Kang

**Affiliations:** 1College of Mining Engineering, Taiyuan University of Technology, Taiyuan 030024, China; tywk2008@163.com; 2Key Laboratory of In-Situ Property-Improving Mining of the Ministry of Education, Taiyuan University of Technology, Taiyuan 030024, China; kangtianhe@163.com

**Keywords:** doped, CH_4_, adsorption, diffusion, molecular simulation

## Abstract

Because kaolinite includes a large range of defect elements, the effects of Mg, Fe(II), and Al doping on the CH_4_ adsorption and diffusion on the surface of Na-kaolinite (001) were investigated by molecular simulations. The simulation results illustrate that ion doping can significantly reduce the amount of CH_4_ adsorbed by kaolinite, but the type of doped ions has little effect on the amount of adsorption. The specific surface area of kaolinite and the interaction energy between CH_4_ and the kaolinite’s surface are two key factors that can determine CH_4_ adsorption capacity. The first peak value of the radial distribution functions (RDFs) between CH_4_ and the pure kaolinite is larger than that between Mg-, Fe(II)-, and Al-doped kaolinite, which indicates that ion doping can reduce the strength of the interactions between CH_4_ and the kaolinite’s surface. Besides hydrogen and oxygen atoms, interlayer sodium ions are also strong adsorption sites for CH_4_ and lead to a weakened interaction between CH_4_ and the kaolinite’s surface, as well as a decrease in CH_4_ adsorption. Contrary to the adsorption results, ion doping facilitates the diffusion of CH_4_, which is beneficial for actual shale gas extraction.

## 1. Introduction

Presently, as a promising clean energy source, shale gas has become a global research hot spot. [1,2,3]. The United States has 7.2 billion cubic meters of shale gas, while China has 3.6 billion cubic meters and Mexico has about 15.4 trillion cubic meters [4]. Shale is composed of organic matter and inorganic matter. In addition to organic matter, clay minerals have also proven to be another major factor in determining CH_4_ adsorption and diffusion [5]. Therefore, studying the interactions between CH_4_ and clay minerals is necessary for researchers in the field of shale gas.

Clay minerals include many types, such as montmorillonites, illites, kaolinites, etc. In recent years, due to China having richest kaolinite content worldwide [6,7], some studies have been conducted on kaolinite’s interactions with CH_4_. Ji et al. [8] experimentally investigated the sorption isotherms, heat of adsorption, and standard entropy of CH_4_ in kaolinite and found that the CH_4_ sorption isotherms at different temperatures can be fitted well by the Langmuir function; the heat of adsorption is 9.6 kJ/mol, and the standard entropy of CH_4_ in kaolinite is −65.3 J/mol. Liu et al. [9] studied the CH_4_ adsorption capacity and mechanism of kaolinite at 60 °C and at pressures up to 18.0 MPa, finding that the Langmuir maximum adsorption capacity was 3.88 cm^3^/g. In addition to experiments, Zhang et al. [10,11,12] used a molecular simulation to systematically study the interactions between CH_4_ and kaolinite and explained the mechanism of CH_4_ adsorption and diffusion from a microscopic perspective.

However, as indicated by Hinckley, with indexes ranging from 1.44 to 0.18, nature kaolinite includes a large range of defect elements [13]. The primary contaminant cations are Mg, Fe(II), and Al, whose concentration ranges are 0.07–0.71 wt%, 0.14–0.54 wt%, and 0.07–0.31 wt%, respectively [14]. Molecular simulation is an effective tool that can determine the adsorption properties of complex and complicated systems under a few extreme conditions [15], such as the influence of cations where experiments are difficult to implement. Mignon et al. [16] used molecular dynamics to study the influence of isomorphic substitution on the hydration/adsorption behavior during the swelling process of montmorillonite. However, to our knowledge, the influences of Mg, Fe(II), and Al doping on the interaction between CH_4_ and kaolinite have not yet been determined.

Thus, in this paper, the molecular simulation method was utilized to investigate the effects of Mg doping, Fe(II) doping, and Al doping on CH_4_ adsorption and diffusion on Na-kaolinite’s (001) surface. We expected that this research will not only offer a deeper understanding of the adsorption and diffusion properties of the pure, Mg-doped, Fe(II)-doped, and Al-doped kaolinite but will also serve as a foundation for further studies on CH_4_ storage and exploration in shale reservoirs.

## 2. Simulation Details

### 2.1. Models

Kaolinite is one of the most abundant clay minerals in the world. A structural model and supercell model of kaolinite, as well as an all-atom model of CH_4,_ were successfully built and applied in a previous study [17], as shown in Figure 1.

Based on the supercell model of kaolinite and the nature isomorphic substitution in kaolinite, doped kaolinite models, where the Mg, Fe(II), and Al concentrations are as low as 0.184 mol%, were built in two different ways. To build the Mg-, Ca-doped kaolinite models, we substituted 1 Al^3+^ with Mg^2+^/Fe^2+^ every 64 Al^3+^ in the aluminum octahedron layer in the same location [18]. However, to build the Al-doped kaolinite model, we substituted 1 Si^4+^ with Al^3+^ every 64 Si^4+^ in the silicon–oxygen tetrahedron layer [19]. The negative charge caused by lattice substitution is compensated by interlayer Na^+^ [20].

### 2.2. Grand Canonical Monte Carlo Simulation

The Grand Canonical Monte Carlo simulation (GCMC) has been identified as one of the most effective ways to study gas adsorption behaviors [21,22]. Therefore, GCMC also used in this article. In CGMC, fugacity is often calculated by the Peng–Robinson equation of state to reflect the effective pressure of the gas [23]. The Lennard–Jones equation, with a cut-off radius of 0.72 nm [24,25], was used to calculate the van der Waals interactions between the kaolinite atoms and CH_4_. The long-range electrostatic interactions and Coulomb interactions were both obtained by three-dimensional Ewald methods, with the accuracy of 4.186 × 10^3^ kJ/mol [26]. The Dreiding force field [27] was utilized to reflect the atoms’ LJ parameters and charges. Moreover, we applied energy changes to decide whether the CH_4_ molecule is accepted, rejected, or displaced in the output via the Metropolis arithmetic rule [28]. The periodic boundary condition was selected for operation. Then, for each study, a total of 1 × 10^7^ configurations were output, half of which were utilized for the system to reach equilibrium, while the other half were utilized for production. Finally, these configurations were applied as the initial models for molecular dynamics (MD) simulations [29].

### 2.3. Molecular Dynamic Simulation

First, the initial configurations of the MD simulations (the final configurations of the GCMC simulation) are minimized by the conjugate gradient and the steepest descent method [21]. Then, 5 ns NVT (T = 293.15 K) MD simulations (1.0 fs time step) were used to reach an equilibrium state. Finally, 5 ns NPT MD simulations (1.0 fs time step) were used to calculate the diffusion coefficients and radial distribution function (RDF). These MD simulations were also applied according to the Dreiding force field and the Velocity Verlet algorithm integrations [30]. The above simulations all conducted by the Sorption and Forcite modules of the Materials Studio package [31].

## 3. Results and Discussion

We compared the results of our previous studies with experimental data from multiple simulation factors, such as adsorption isotherms, lattice vectors, pore volumes, and diffusion coefficients [11,12]. These data verify that the force field and model utilized in this paper can continue to be used to study the adsorption and diffusion of CH_4_ in kaolinite by ion doping.

### 3.1. CH_4_ Adsorption Amount

Figure 2 shows the absolute adsorption amount of CH_4_ on pure and Mg, Fe(II), and Al-doped kaolinite at a temperature of 293.15 K up to 20 MPa. The results illustrate that, under the same pressure and temperature conditions, ion doping can significantly reduce the amount of CH_4_ adsorbed by kaolinite. However, the type of doped ions has little effect on the amount of adsorption. In order to fit the absolute rate, the Langmuir model [32] was utilized to analyze the simulated rate. The Langmuir model can be displayed as follows:(1)$$N=\frac{N_{L}P}{\left(P_{L}+P\right)}$$
where *N* is the absolute adsorption capacity of the CH_4_ in molecules/unit cell; *N_L_* is the Maximum adsorption capacity of the CH_4_ in molecules/unit cell; *P_L_* is the Langmuir pressure (MPa); and *P* is the present pressure in MPa. This model has been identified as one of the most effective ways to study gas adsorption behaviors, from macroscopic to microscopic [33].

Table 1 lists the Langmuir constants obtained by fitting the Langmuir model to the absolute adsorption amount of CH_4_ on pure and Mg-, Fe(II)-, and Al-doped kaolinite at 293.15 K. From Table 1, we can see that the saturated adsorption of CH_4_ on the surface of the pure kaolinite is significantly larger than that after doping. However, the type of doped ions has also little effect on the saturated adsorption of CH_4_ on the surface of the kaolinite. In addition, the Langmuir pressure *P_L_* can reflect the affinity of the absorbent for CH_4_ and the feasibility of CH_4_ desorption under the reservoir pressure. A lower PL value indicates that CH_4_ is more easily adsorbed and that desorption is more difficult to achieve. The CH_4_ Langmuir pressure of the doped kaolinite also becomes larger, which indicates that doping ions may improve the desorption efficiency of CH_4_.

### 3.2. Specific Surface Area and Interaction Energy

According to the studies of Frost et al. [34], Sui and Yao [35], Thomas [36], and Zhang et al. [10], the specific surface area and interaction energy can determine the adsorption capacity of the adsorbate on the adsorbent. This is because the former can determine the adsorption space and the latter can determine the adsorption strength. Thus, in this research, we also used these two key parameters to analyze the effect of ion doping on the adsorption mechanism of CH_4_ on the surface of kaolinite.

The Connolly surface area was used to reflect the specific surface area of kaolinite, which has been widely used in molecular simulation studies [37,38]. The Connolly surface area is the area traced out by the contact surface of the probe molecule as it freely rolls over the kaolinite. This is also called the solvent accessible area [39]. We used the non-adsorbed helium gas as the probe molecule, with a van der Waals radius of 1.0 Å.

Figure 3 shows the histogram of the specific surface area of pure and Mg-, Fe(II)-, and Al-doped kaolinite. The data show that ion doping can significantly reduce the specific surface area of kaolinite. The specific surface area of pure kaolinite and Mg-doped, Fe(II)-doped, and Al-doped kaolinite were 2026.37, 1829.12, 1829.13, and 1828.98 m^2^/g, respectively. Compared to pure kaolinite, the specific surface area of the kaolinite after ion doping reduces by approximately 9.69%. Interestingly, the effect of ion type on the specific surface area of the kaolinite is not significant.

The interaction energies between CH_4_ and pure and Mg-, Fe(II)-, and Al-doped kaolinite were obtained by
(2)$$E_{\mathrm{int}eraction}=E_{CH_{4}-kaolinite}-(E_{CH_{4}}+E_{kaolinite}).$$
where E_CH4__−kaolinite_ is the energy of CH_4_ and the pure and Mg-, Fe(II)-, and Al-doped kaolinite complex; E_CH4_ and E_kaolinite_ are the energies of CH_4_ and the pure and Mg-, Fe(II)-, and Al-doped kaolinite isolates, which were obtained from a single point [40,41]. Figure 4 displays the interaction energies of CH_4_ adsorption on pure and Mg-, Fe(II)-, and Al-doped kaolinite at different pressures. The results show that the interaction energy of CH_4_ adsorption on pure and doped kaolinite all increased logarithmically with increasing pressure according to *E* = 2.0398ln(*p*) + 36.274. Under the same pressure conditions, the interaction energy between CH_4_ and pure kaolinite is larger than the interaction energy between CH_4_ and Mg-, Fe(II)-, and Al-doped kaolinite. In addition to the specific surface area, this is another reason CH_4_ adsorbs more on the surface of pure kaolinite than on the surface of doped kaolinite. In terms of the interaction energy with CH_4_, pure kaolinite > Mg-doped kaolinite > Fe(II)-doped kaolinite > Al-doped kaolinite. This observation is consistent with the CH_4_ adsorption shown in Figure 2.

### 3.3. Adsorption Strength and Adsorption Site

There are numerous studies that have successfully used radial distribution functions (RDFs) to analyze the adsorption strength and adsorption sites of adsorbates on adsorbents [42,43,44,45]. In this study, RDFs were used to analyze the adsorption strength and adsorption site between CH_4_ and atoms of pure and doped kaolinite.

Figure 5 shows the radial distribution functions (RDFs) between CH_4_ and the pure, Mg-, Fe(II)-, and Al-doped kaolinite surface at 293.15 K and 20 MPa. For greater clarity, the first peak portion has been partially enlarged. The results show that the first peak values are 1.35, 1.33, 1.31, and 1.27 with an adsorption distance of 3.624, 4.375, 4.625, and 4.875 Å, corresponding to the pure, Mg-doped, Fe(II), and Al-doped kaolinite surface, respectively. This illustrates that it is 1.35, 1.32, 1.31, and 1.30 times more likely that CH_4_ would be adsorbed at this separation. The first peak value between CH_4_ and the pure kaolinite is larger than that between the Mg-, Fe(II)-, and Al-doped kaolinite, which indicates that ion doping can reduce the strength of the interactions between the CH_4_ and kaolinite surfaces. This is consistent with the conclusions in Figure 4. There are some uncertainties in the simulation results, which are mainly related to the hardness of the potential repulsion nucleus between the kaolinite molecules and the CH_4_ molecules [46].

Figure 6 displays the RDFs between the CH_4_ and the different atoms of pure and doped kaolinite. The results show that the first peaks of RDFs between the CH_4_ and H atoms and between CH_4_ and O atoms are sharper and stronger than those between the CH_4_ and Si atoms, between the CH_4_ and Al atoms, between the CH_4_ and Mg atoms, and between the CH_4_ and Fe(II) atoms in the kaolinite. This illustrates that the hydrogen and oxygen atoms of kaolinite are a strong adsorption site for CH_4_, which is consistent with the results of our previous study [11]. Interestingly, in addition to hydrogen and oxygen atoms, CH_4_ can strongly adsorb at the sites of the sodium atoms between the two surfaces of kaolinite, as shown in Figure 7. Figure 7 shows snapshots of CH_4_ adsorption in pure, Mg-doped, Fe(II)–doped, and Al-doped kaolinite. It can be clearly seen that, after ion doping, CH_4_ is strongly adsorbed around sodium ions and has a larger pore size change. After doping, the overall structure of kaolinite will be negatively charged, so sodium ions are used for compensation [20]. Thus, the structural formula of kaolinite has changed, which can cause an increase in the basal spacing of kaolinite [47,48]. This will lead to weakened interactions between the CH_4_ and kaolinite surfaces and a decrease in CH_4_ adsorption.

### 3.4. CH_4_ diffusion Coefficients

The Einstein diffusion law [49] was used to calculate the CH_4_ diffusion coefficients on the pure and Mg-, Fe(II)-, and Al-doped kaolinite by averaging the mean squared displacement (MSD) of a single molecule over time:(3)$$D_{s}=\frac{1}{6N}\lim_{t\to \infty }\frac{d}{d_{t}}\langle \sum_{t=1}^{N}{\left[r_{i}\left(t\right)-r_{0}\left(t\right)\right]}^{2}\rangle .$$
where denotes the number of the overall average properties, *N* is the amount of diffusion molecules, and *r_i_* (*t*) is the distance vector of the *i*th molecule at time *t*.

Figure 8 displays the self-diffusion coefficients of CH_4_ on pure and Mg, Fe(II), and Al-doped kaolinite at different pressures. With an increase in pressure, the diffusion coefficient decreases sharply and then increases slowly. This is the result of a combination of mechanical compression and swelling caused by the adsorption of CH_4_ [50]. It is interesting that the diffusion coefficient of CH_4_ after doping is larger than that of pure kaolinite. This is due to the fact that ion doping can increase the pore size of kaolinite, which can further affect the CH_4_ diffusion coefficient. The CH_4_ diffusion coefficient increased exponentially with the pore size [12]. This means that ion doping facilitates the diffusion of CH_4_, which is beneficial for actual shale gas extraction.

## 4. Conclusions

Molecular simulations were utilized to investigate the CH_4_ adsorption and diffusion in pure and Mg-, Fe(II)-, and Al-doped kaolinite. The results demonstrate that the saturated adsorption of CH_4_ on the surface of pure kaolinite is significantly larger than that after doping. In terms of the relative CH_4_ sorption capacity, pure kaolinite > Mg-doped kaolinite > Fe(II)-doped kaolinite > Al-doped kaolinite. Compared with pure kaolinite, the specific surface area of kaolinite after ion doping is reduced by approximately 9.69%. The interaction energies of CH_4_ adsorption on the pure and doped kaolinite all increased logarithmically with an increase in pressure according to *E* = 2.0398ln(*p*) + 36.274. The first peak value between CH_4_ and the pure kaolinite is larger than that between the Mg-, Fe(II)-, and Al-doped kaolinite, which indicates that ion doping can reduce the strength of the interactions between CH_4_ and the kaolinite’s surface. The diffusion coefficient of CH_4_ after doping is larger than that of pure kaolinite. This means that ion doping facilitates the diffusion of CH_4_, which is beneficial for actual shale gas extraction. Molecular simulations are thus an efficient means to investigate the influence of ion doping on gas adsorption and diffusion characteristics.

## Figures and Tables

**Figure 1 molecules-25-01001-f001:**
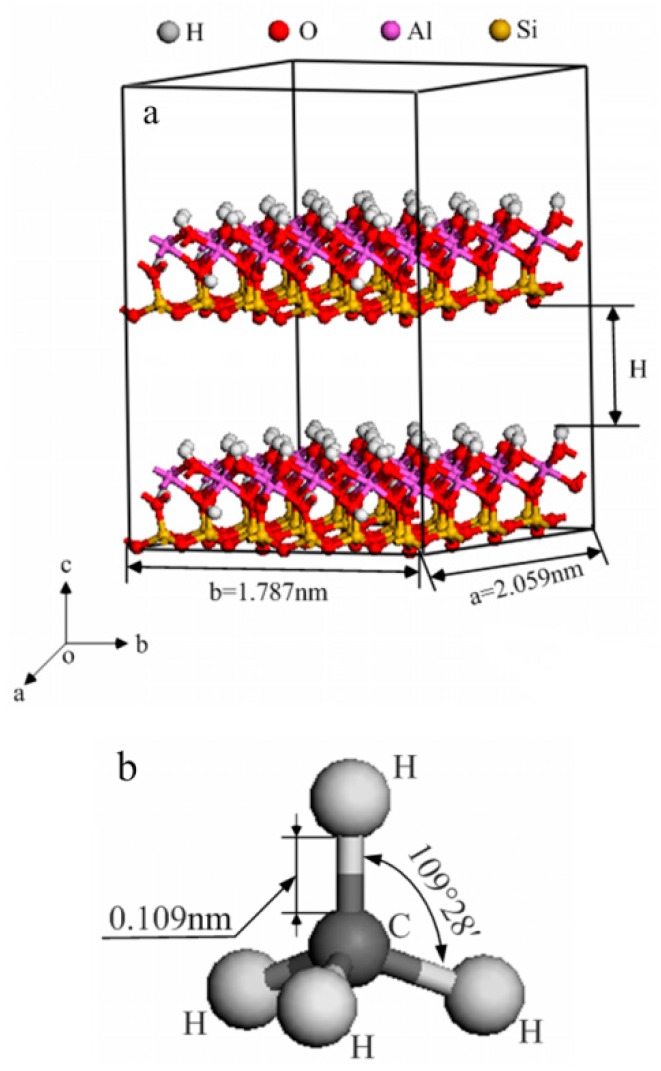
The 4 × 2 × 2 kaolinite (001) surface super model (**a**) and CH_4_ molecular model (**b**) [17].

**Figure 2 molecules-25-01001-f002:**
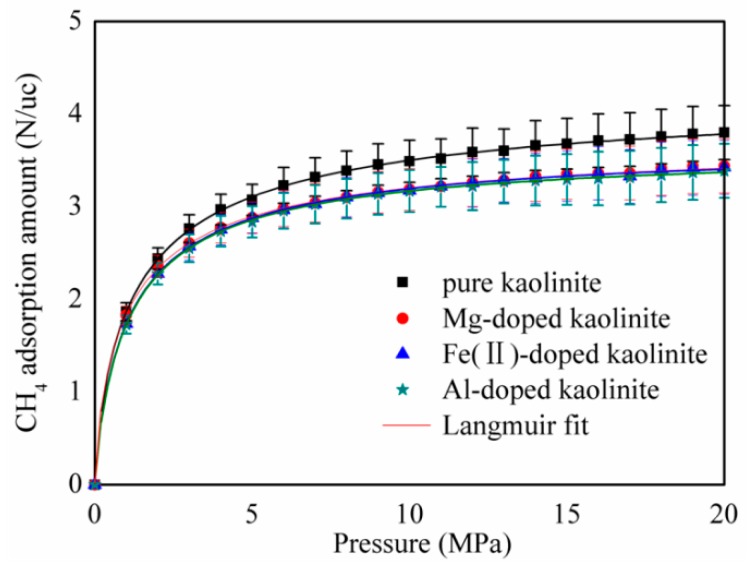
Absolute adsorption amount of CH_4_ on pure and Mg-, Fe(II)-, and Al-doped kaolinite at a temperature of 293.15 K. Error bars indicate the standard deviations.

**Figure 3 molecules-25-01001-f003:**
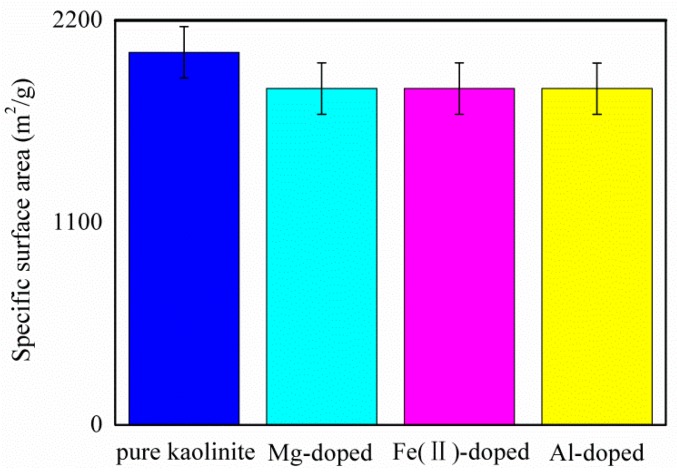
The histogram of the specific surface area of pure and Mg-, Fe(II)-, and Al-doped kaolinite. Error bars indicate the standard deviations.

**Figure 4 molecules-25-01001-f004:**
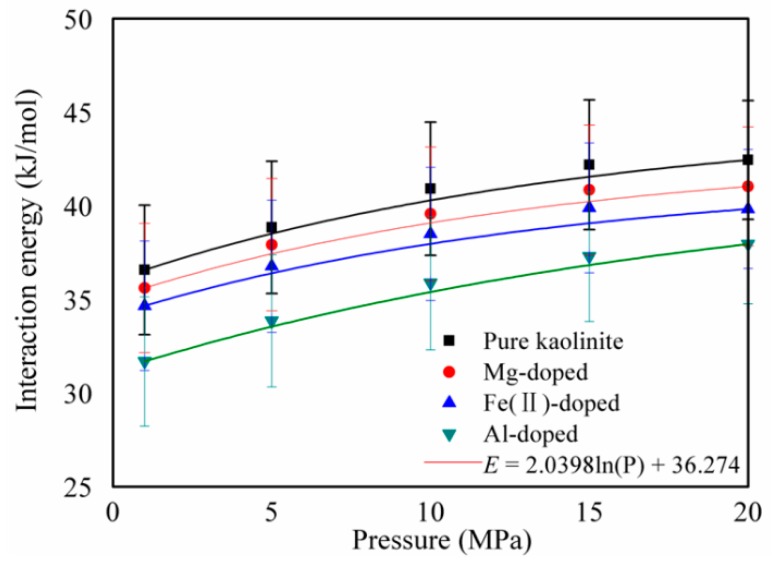
Interaction energies of CH_4_ adsorption on pure and Mg-, Fe(II)-, and Al-doped kaolinite at different pressures. Error bars indicate the standard deviations.

**Figure 5 molecules-25-01001-f005:**
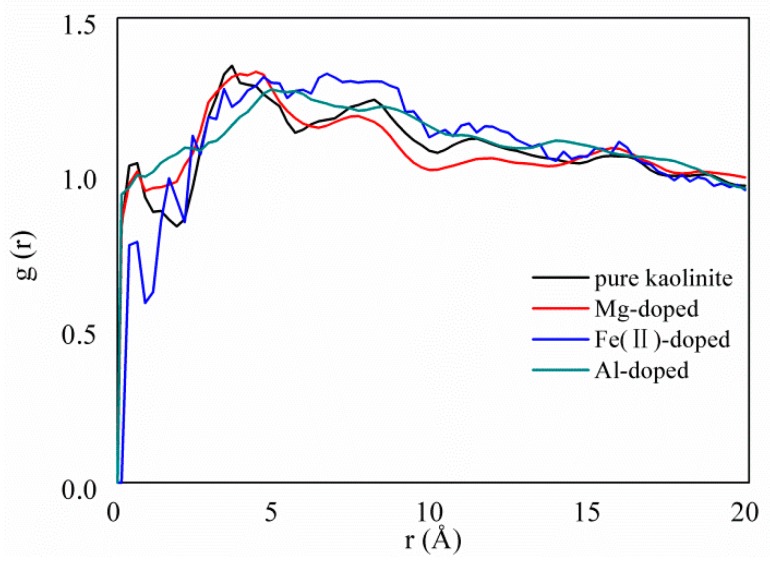
Radial distribution functions (RDFs) between the CH_4_ and the pure, Mg-, Fe(II)-, and Al-doped kaolinite surface at 293.15 K and 20 MPa.

**Figure 6 molecules-25-01001-f006:**
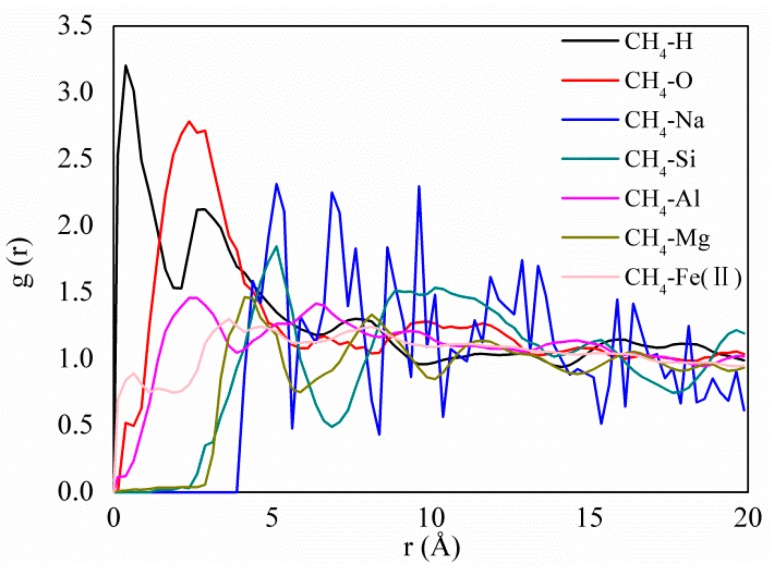
RDFs between the CH_4_ and different atoms of pure and doped kaolinite.

**Figure 7 molecules-25-01001-f007:**
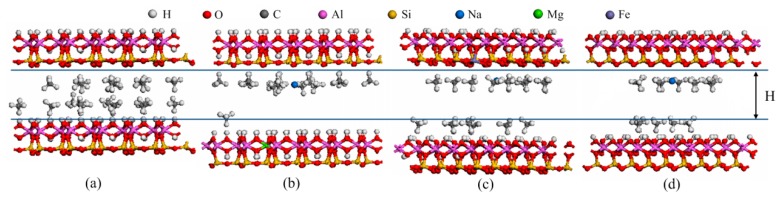
The snapshots of CH_4_ adsorption. (**a**) pure kaolinite; (**b**) Mg-doped kaolinite; (**c**) Fe(II)-doped kaolinite; (**d**) Al-doped kaolinite.

**Figure 8 molecules-25-01001-f008:**
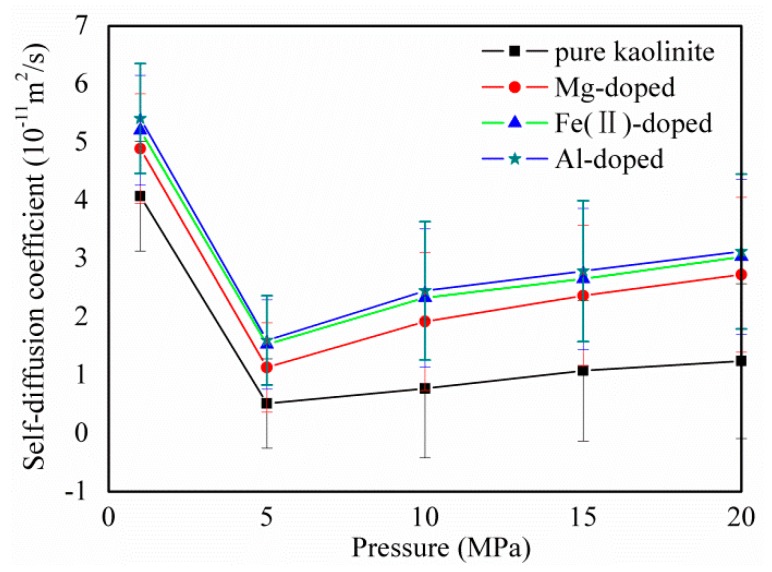
The self-diffusion coefficients of CH_4_ on pure and Mg-, Fe(II)-, and Al-doped kaolinite at different pressures. Error bars indicate the standard deviations.

**Table 1 molecules-25-01001-t001:** Langmuir constants obtained by fitting the Langmuir model to the absolute adsorption amount of CH_4_ on pure and Mg-, Fe(II)-, and Al-doped kaolinite at 293.15 K.

Surface	*V_L_* (molecules/uc)	*P_L_* (MPa)	Correlation Coefficient R^2^
Kaolinite (001)	3.91	0.95	0.992
Mg-doped	3.54	1.01	0.991
Fe(II)-doped	3.51	1.05	0.996
Al-doped	3.50	1.06	0.997

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
