# Peer review of "The Effect of Mg, Fe(II), and Al Doping on CH4: Adsorption and Diffusion on the Surface of Na-Kaolinite (001) by Molecular Simulations"

_molecules, 2020, doi:10.3390/molecules25041001_

Round 1

Reviewer 1 Report

The authors present GCMC and MD simulation results for the adsorption and diffusion of CH4 on pure kaolinite in comparison with Mg, Fe and Al doped surfaces. This is generally an interesting work. However, important simulation details are missing, therefore it is not possible to judge the quality and reliability of the simulation results. Thus, I recommend publication in Molecules only after a major revision.

I suggest that the authors address the following points in the revision of the manuscript:

In line 73, the authors provide the cut-off radius for the Lennard Jones interactions (0.72) - though without unit. I assume it is 0.72 nm = 7.2 Å? That would be a small cut-off radius, and the authors should comment if they applied any long range corrections. In line 78 they mention that they employed the Ewald sum - but without providing details on its parameters/settings. In line 79-80, the authors state that “in order to reach equilibrium and obtain adsorption amount of CH4, 2*107 configurations were generated”. How many configurations were needed to be generated to reach equilibrium, how many different configurations were generated in the production run of the simulation? How did they generate different configurations? Only by insertion and deletion trial moves, or also by displacements of adsorbed CH4 molecules etc? How did they determine that equilibrium was reached? In the MD simulations (Section 2.3), how did they ensure that the simulated temperature is 293 K? Prior to the simulation in the NVE ensemble they should have performed an equilibration in the NVT ensemble. In the Section 2.3, the authors to not provide any details on their molecular dynamics simulations: what is the time step, which integrator did they use etc. What is the length of the MD simulations? The very edgy RDFs shown in Figures 5 & 6 suggest that the lengths of the MD simulations have not been long enough to yield reliable results (and smooth RDF curves). The authors do not provide any uncertainties of their molecular simulation results. However, molecular simulation results without states uncertainties are meaningless. Thus, the manuscript cannot be accepted for publication unless the authors provide a statistical analysis of their simulation results. In the equation given in line 138, units are again missing. It is not clear how the authors determined the specific area and interaction energy (section 3.2) or the pore size and volume change (line 173).

Reviewer 2 Report

In this paper, authors described the adsorption of CH4 into Kaolinite channels.

The design of the experiment is appropriate, but some checks should be done. Moreover, some English style errors should be corrected.

Both in the Introduction and in the Methods section, the abundance of clay minerals in China are given, but this is an international journal and other data on the diffusion of this clays overall the world should be given. Please, do not take into account only China. How do authors substitute the doping atom into the clay? In random positions or in the same location for all the models? Does the location of the doping agent affect the results such as the adsorption energy? How many atoms of CH4 are inserted?  Add a reference for Material Studio package "It can be clearly seen that the ion doping can significantly reduce the specific surface area of kaolinite": how is the area calculated? Is the accessible area? "Interestingly, the effect of ion type on specific surface area of kaolinite is not significant": does this depends on where is the doping agent? "There are numerous studies used the radial distribution.." correct into "... numerous studies that uses..." " and adsorption site of adsorbate on adsorbent successfully[...]": for this sentence consider also doi.org/10.1021/acs.jpcc.6b09983 Line 165: The results showS...   Line 177: Figure 7 displayS... From Figure 7 it seems that the pore size increases when using a dopant. It is true? Do authors have any explanation of this effect?  Moreover, the CH4 molecules seems to be well ordered in the case of Al-doped. Why?  The diffusion coefficient is affected by the pore size of kaolinite. Can authors discuss about this? Finally, in Conclusion section, please do not repeat all results, but only the most significative so the reader ca understand the relevance of this experiment.

Round 2

Reviewer 1 Report

The authors have made an effort to comply with the recommendations of the reviewers, and to make appropriate changes to the manuscript. However, they have ignored my main critism – i.e. point (5) of my first review:

The authors do not provide any uncertainties of their molecular simulation results. However, molecular simulation results without states uncertainties are meaningless. Thus, the manuscript cannot be accepted for publication unless the authors provide a statistical analysis of their simulation results.

That means that the authors need to show the errors bars of their simulation results in Figures 1, 3, 4 and 7. Determining the statistical errors is not so difficult to do – the authors only need to perform several independent simulation runs and statistically evaluate the different simulation results. This is explained in every textbook on molecular simulations i.e. by

Frenkel & Smit: Understanding Molecular Simulation

Allan & Tildesley: Computer Simulation of Liquids

Raabe: Molecular Simulation Studies on Thermophysical Properties.

The RDFs shown in Figures 5 & 6 are still very edgy, and therefore not sound. The authors have probably chosen the width of the dr bins much too large. Therefore there should repeat the evaluation with smaller dr value in order to obtain meaningful RDFs.

Reviewer 2 Report

Authors replied to all my answers improving the quality of the manuscript. Methods are correctly described and conclusion are supported by simulation results.

Author Response

Thank you very much for your previous questions and suggestions.